# Optimizing mouse metatranscriptome profiling by selective removal of redundant nucleic acid sequences

Morgan Roos,[1] Samuel Bunga,[1] Asako Tan,[1] Erica Maissy,[2] Dylan Skola,[1] Alexander Richter,[2] Daniel S. Whittaker,[3] Paula Desplats,[4] Amir Zarrinpar,[2,5,6,7] Rick Conrad,[8] Scott Kuersten[1]

ABSTRACT  Metatranscriptome (MetaT) sequencing is a critical tool for profiling the dynamic metabolic functions of microbiomes. In addition to taxonomic information, MetaT also provides real-time gene expression data of both host and microbial populations, thus permitting authentic quantification of the functional (enzymatic) output of the microbiome and its host. The main challenge to effective and accurate MetaT analysis is the removal of highly abundant rRNA transcripts from these complex mixtures of microbes, which can number in the thousands of individual species. Regardless of the methodology for rRNA depletion, the design of rRNA removal probes based solely upon the taxonomic content of the microbiome typically requires very large numbers of individual probes, making this approach complex to commercially manufacture, costly, and frequently technically infeasible. In previous work (A. Tan, S. Murugapiran, A. Mikalauskas, J. Koble, et al., BMC Microbiol 23:299, 2023, https://doi.org/10.1186/s12866-023-03037-y), we designed a set of depletion probes for human stool samples using a design strategy based solely on sequence abundance, completely agnostic of the microbiomal species present. Here, we show that human-based probes are less effective when used with mouse cecal samples. However, adapting additional rRNA depletion probes specifically to cecal content provides both greater efficiency and consistency for MetaT analysis of mouse samples.

IMPORTANCE  Sequencing total RNA from microbiome samples is seriously impaired by the overwhelming proportion of rRNA to mRNA content. As much as 99% of sequencing reads can be assigned to the rRNA content, thus removal of these abundant transcripts is critical to metatranscriptome (MetaT) analysis. The use of Ribo Zero Plus rRNA depletion probes designed for human gut microbiomes proved to be less effective and more inconsistent across mouse cecal donor samples, a common experimental system for microbiome studies. In the present work, we have extended and refined a taxonomically neutral probe design method for mouse cecal content. The additional probes were carefully chosen to limit the number needed for effective depletion to reduce both the cost and risk of introducing bias to MetaT analysis. Our results demonstrate this method as efficient and consistent for rRNA removal in mouse cecal samples, thus providing a significant increase in the number of mRNA-rich sequencing reads for MetaT analysis.

KEYWORDS  nonhuman microbiome, metatranscriptome, rRNA, gene sequencing, mRNA

The diversity and metabolic state of the estimated ~1,000 species contributing to the intestinal microbiota are immensely important to the health and well-being of the host (1). Diet, overall health, and even disease state of the host can, conversely, affect the activity and enzymatic expression patterns of the intestinal microbiome. Our understanding of microbiomes has evolved over the past decades, such that it is

Editor Aaron W. Miller, Cleveland Clinic, Cleveland, Ohio, USA

Peer Reviewer Mangesh Suryavanshi, Cleveland Clinic, Cleveland, Ohio, USA

Address correspondence to Scott Kuersten, skuersten@illumina.com.

Morgan Roos and Samuel Bunga contributed equally to this article. The order of these authors was determined based upon relative contribution.

The authors declare no conflict of interest.

now understood that the interplay between the microbial population and its host is a dynamic and vital interaction (2). To further understand this interaction, it is crucial that we look at not just the composition of the microbes present but also the metabolic contribution from this population to assess the health and well-being of the host.

In the last decades, the use of next-generation sequencing (NGS) to interrogate microbiomes, especially from the gut contents of host organisms, has become more cost-effective and technologically attainable (3, 4). Genetic profiling of the microbial populations present in gut microbiomes is traditionally performed by methods such as 16S rDNA sequencing (5–7). More recently, as NGS has become far less expensive, whole genome bacterial shotgun sequencing has become more commonplace and provides a greater depth of knowledge of the bacterial diversity and inferred function(s) in these complex communities (8–10). However, metagenome (MetaG) sequencing is mostly limited to taxonomic identification of genus and/or species composition of the samples. The functional metabolic state of the microbes can, at best, be deduced based on assumptions of the relative contributions from the taxonomic groups present. Elucidating the profile of mRNAs being expressed by these populations (their transcriptomes) is more informative.

Using transcriptome profiling of microbiomes (i.e., metatranscriptomics) can provide a wealth of valuable information about not just the taxonomic composition, but also the metabolic activity of the microbial population (11). The dynamics of gene expression changes between microbiome and host can be simultaneously monitored to establish the health or disease states (12, 13). Metatranscriptome (MetaT) sequencing offers a much more detailed view of the overall activity of the microbes by providing real-time functional gene expression information, such as what gene families and enzymatic activities are taking place at the time of sample collection and extraction (14–17). Furthermore, it can be quite informative to track these dynamic changes over time, especially in response to dietary interventions, drug treatments, and disease progression of the host (1, 18). Whereas tracking changes to the taxonomic composition of the gut by MetaG sequencing may be relatively stable upon intervention, MetaT offers the potential for a more detailed and dynamic profiling of the most transcriptionally active participants in the sample (19).

A major challenge for MetaT analysis is the presence of highly abundant rRNA transcripts (16, 20, 21). Bacterial small subunit (SSU or 16S) and large subunit (LSU or 23S) rRNA transcripts dominate total RNA samples extracted from gut microbiomes. Indeed, sequencing total RNA from stool samples without rRNA removal can typically result in >95% of the reads matching LSU and SSU rRNA transcripts. Methods to remove the vast diversity of microbiome rRNA can be contrasted with the relatively easy methods available for removing host eukaryotic rRNA from samples. Since polyA tails are added post-transcriptionally to coding mRNAs as part of eukaryotic RNA processing, the mRNAs themselves can be preferentially enriched using oligo-dT capture beads. Additionally, removal of the host rRNA is more feasible since it only requires using sequences from a single species. Several commercially available rRNA removal kits and methods can be utilized to remove rRNA from commonly studied species, such as human, mouse, and rat, and provide the means to sequence both mRNA and non-coding transcripts in these sample types. Routinely used methods for removing rRNA from total RNA include enzymatic (i.e., RNase H) depletion, CRISPR-based approaches, or physical removal using hybridization with antisense biotinylated probes and streptavidin magnetic beads (20, 22–24). However, these methods designed for samples from a single eukaryotic host are not effective for complex microbiome sample types since the microbial rRNA is from a multitude of source organisms whose sequences are evolutionarily divergent (25). Removal of these abundant transcripts through targeted probe design, whether using physical or enzymatic means, is a much more complex procedure to permit the deep sequencing of microbial mRNA.

In a recent study, a "rational" probe design strategy was established where raw sequencing data were utilized to collect and filter abundant sequences from human

gut microbiome samples (25). The probe set that was designed demonstrated effective and efficient enzymatic depletion of rRNA from both human adult and infant stool samples with minimal bias introduced. Robust depletion of rRNA within multiple human microbiome sample types, including stool, tongue, and vagina, resulted in >60% of sequencing reads available for MetaT analysis. This method is commercially available as the Ribo-Zero Plus Microbiome (RZPM) kit, which is an enzyme-based (RNase H) depletion method. This method also relies upon DNase to eliminate the DNA probes following depletion; therefore, a limitation of this technique is the potential of including an excess of probes that effectively saturate DNase and contaminate the subsequent libraries. Probe design based solely upon the large number of taxa present in gut microbiomes would require far too many oligonucleotides for the method to process. Furthermore, the conversion of the highly structured, GC-rich rRNA into cDNA is inconsistent due to the inherent limitations of reverse transcriptase processivity, so not all rRNA sequences are abundantly represented in the libraries, and inclusion of probes to these sequences is not needed. However, probe design based upon the most abundant rRNA sequences present in the samples is a more rational approach because it both limits the number of probes needed for effective depletion and reduces the overall cost of the assay.

In this work, we sought to first test the ability of RZPM to deplete rRNA from a set of mouse microbiome samples and, if needed, make use of a similar rational probe design strategy to create supplemental probe pools optimized for mouse gut (specifically, cecal) contents. Since there is currently no commercially available solution for rRNA depletion of mouse microbiome sample types, an important goal is to provide a list of additional probes that can be obtained at minimal cost and combined with RZPM for more effective performance. Additionally, this effort is intended to provide improvements in both consistency and efficiency of depletion for mouse-specific MetaT analysis. We demonstrate that the addition of the supplemental probes can provide ~75% of the mRNA-rich reads available for MetaT analysis. These extra probes add a minimal cost per sample to the existing RZPM kit, yet provide an additional ~15% of sequencing reads for functional data analysis.

## MATERIALS AND METHODS

### Animal work

Cecal contents were manually removed (not washed) from the cecum and saved for RNA extractions. For the terminal ileum, the distal ~2 cm of the ileum was collected, including the ileal content. Samples were immediately flash frozen and powderized, and RNA was extracted using the MagMAX mirVana Total RNA isolation kit (Thermo Fisher #A27828).

### RNA-Seq library preparation

Extracted RNA (~100–200 ng) was prepared for sequencing using the Illumina Stranded Total RNA Prep with Ribo-Zero Plus Microbiome kit following the manufacturer's protocol. The 5050 and 2025 probe sets were purchased from IDT as an oPools product at 50 pmol per oligo. For each tube, the lyophilized pellet obtained from the manufacturer was resuspended in 50 µL of nuclease-free water (for a final concentration of 1 pmol/µL/oligo, which matches the concentration of probes contained in the Ribo-Zero Plus Microbiome kit), and 1 µL was used per depletion reaction. Ribo-Zero Plus depletion was performed with either (i) no probes (volume supplemented with nuclease-free water), (ii) Depletion Pool 1 (DP1) and Depletion Pool Microbiome (DPM) (1 µL each), (iii) DP1, DPM, and the 5050 probe set (1 µL each), or (iv) DP1, DPM, 2025 probe set (1 µL each). RNA-Seq libraries were sequenced on Illumina NovaSeq 6000 at 2 × 150 bp.

## RNA-Seq analysis

Post-sequencing, FASTQs were downsampled to 20 million total reads (10 million clusters) each using BaseSpace Sequencing Hub (BSSH) DRAGEN FASTQ Toolkit application and then processed further with additional BSSH applications. For meta-transcriptome analysis, the Microbiome Metatranscriptomics application was used. The BBDUK output from this application was utilized for analysis of rRNA content and for rRNA-based taxonomic analysis. As part of filtering out the rRNA reads, the application bins the rRNA reads into a FASTQ file for each sample. The rRNA reads were then processed with the BSSH DRAGEN Metagenomics application for species analysis and proportion. For host transcriptome and differential gene expression analysis, the BSSH RNA Express application was used. For normalization of gene family abundance, HUMAnN3 (contained within the Microbiome Metatranscriptomic App) quantifies genes and pathways in units of reads per kilobase. To make the results comparable across samples, we converted these raw abundances to relative abundances using the human renorm table script with the -u relab parameter (as described in the HUMAnN3 documentation at https://github.com/biobakery/humann). This process follows a total sum scaling approach, which normalizes the total abundance in each sample to 1, which helps control for differences in sequencing depth (26). Table S3 contains a list of the BSSH Apps used for analysis of this work.

## Probe design

Following sequencing and RNA-Seq analysis, seven cecal samples with more than 30% of reads aligning to rRNA were identified. The rRNA reads were collected and processed with variations of a probe design pipeline described in previous work (25). To select the most efficient and cost-effective probe counts, the samples were processed with the probe design pipeline multiple times, varying the number of the top-most abundant regions from 20 to 50 and altering the spacing between probes from 25 to 50 nt. Out of the matrix of results, two options were selected based on the number of probes yielded and the level of stringency. These probe sets were then assessed for their performance in library preparation and sequencing. The list of the 2025 and 5050 probe sequences is found in File S1. The scripts used for probe design are available on request.

## RESULTS

### Efficacy of rRNA depletion of mouse cecal samples

In previous work, we demonstrated the utility of using the enzyme-based RiboZero Plus methodology to deplete rRNA from mixed-source RNA extracted from complex human microbiomes (25). We developed a strategy to efficiently design removal probes for complex mixtures of microbes based solely on sequence data without referencing assumptions about specific bacterial genera or species being present. We refer to this as "rational" or "taxonomically neutral" design of probes. This probe set was subsequently commercialized in the Ribo Zero Plus Microbiome kit as DPM. Combining DPM with the human, mouse, and rat probes from the Ribo Zero Kit (DP1) demonstrated effective depletion of both host and microbial rRNA sequences across several human microbiome sample types, and the resulting RNA-Seq libraries did not show significant bias from the additional probes used for depletion (25).

Since the initial design used microbiome sequences from human stool samples as the training set to provide new content, we investigated how it would perform with microbiome samples obtained from the common mouse, *Mus musculus*, a laboratory model system routinely used for studying the gut microbiome. Numerous studies using methods such as 16S rDNA and whole genome shotgun sequencing of murine gut content have been published (18, 27, 28). However, mouse MetaT analysis has been somewhat limited due to the lack of relevant molecular tools to aid in the generation of mRNA-rich RNA for RNA-Seq library preparation (13, 15–17).

To determine the effectiveness of RZPM depletion from mouse microbiomes, matched samples from ~60 individual mice (56 cecal, 62 terminal ileum, and 62 liver samples) were obtained. The functional analysis of these samples is part of a larger study on aging and Alzheimer's that will be discussed in detail elsewhere (E. Maissy, A. Richter, D. Whittaker, P. Desplats, and A. Zarrinpar, unpublished data). For this work, we initially utilized the commercial RZPM probe sets (DP1 + DPM) to determine their effectiveness and the extent of rRNA depletion in these three sample types. Total RNA extracted from each sample was subjected to rRNA depletion using the RZPM kit, converted into RNA-Seq libraries using the Illumina Stranded Total RNA Kit, and sequenced (150 cycle paired end) on a NovaSeq 6000 instrument (Fig. 1A). The data were analyzed using the Microbiome Metatranscriptomics application in BSSH. Briefly, the raw data were quality-filtered and then analyzed for rRNA content by BBDUK alignment using the Silva database, a comprehensive rRNA database, as the alignment reference (25, 29). The pipeline reports the proportion of reads that match rRNA sequences or host (mm9) transcriptome (Fig. 1A). The number of reads that match rRNA sequences differs depending on sample type (Fig. 1B). Both ileum and liver samples have very low rRNA reads from microbiome content and instead are dominated by host transcriptome (mm9). This suggests that these sample types are predominantly composed of mouse cells, and rRNA depletion was effective at removing the confounding host rRNA reads. In contrast, the cecal samples display very little host contribution and instead contain a much higher proportion of microbial ("retained," blue) content and a variable amount of contaminating rRNA reads. This suggests that the cecal samples contain very little mouse RNA and are mostly composed of bacterial content. The proportion of rRNA reads differs for each individual sample and ranges from ~5% to as high as nearly ~50%. This high variability is likely due to differences in total microbial content from each individual mouse and the technical limitations of the RZPM probes designed against human microbiomes to deplete the rRNA content effectively in relatively dissimilar mouse samples (Fig. 1B). If the samples with higher rRNA contamination are the result of a single or even a few specific species, it would be a straightforward and relatively simple task to design new probes directly against the rRNA sequences from those species. However, if the contamination is from many species, probe design would potentially become too cumbersome and would likely benefit from a more taxonomic-free approach. To ascertain the validity of this assumption, the filtered rRNA reads from a subset of both undepleted and depleted cecal samples were collected and processed using the BSSH Dragen Metagenomics Pipeline to confer proportional taxonomic identification to the rRNA content (Fig. 1C). The results suggest that, following depletion, there is not a single or even a few taxa that dominate the remaining leftover rRNA reads in these samples, suggesting the contaminating reads are spread across multiple species. To optimize the performance of rRNA depletion of the cecal samples, we undertook a probe design strategy that is taxonomically neutral and instead relies on the most abundant rRNA sequences in the sample (25).

## Probe design and testing

We took an approach to probe design similar to a strategy used to create a probe set for rRNA removal from human gut microbiome samples (25). The strategy relies on sequence abundance and not on taxonomic information to establish probe sequences. A second consideration is that since Ribo-Zero Plus relies on the use of DNase to remove the probes following enzymatic depletion, too many extra probes could saturate the ability of DNase to digest them. This could potentially lead to excess oligonucleotides being incorporated into the library preparation and result in undesired sequences contaminating the analysis results. For the current set of studies, we therefore examined approaches to reduce the quantity of probes being used for each depletion (Fig. 2). Effective spacing, filtering, and alignment during probe design minimize both the risk of DNase saturation and cost per experiment, which is especially relevant for studies using model organisms like mouse where potentially large numbers of samples are processed.

A

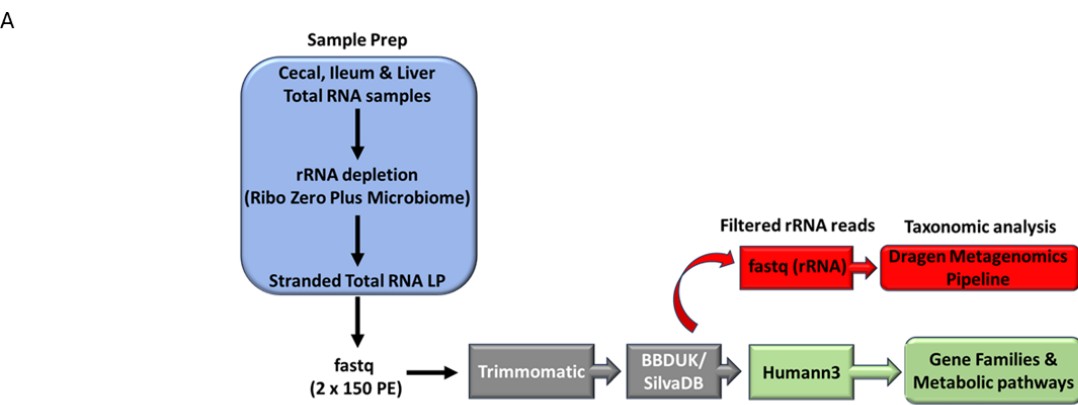

B

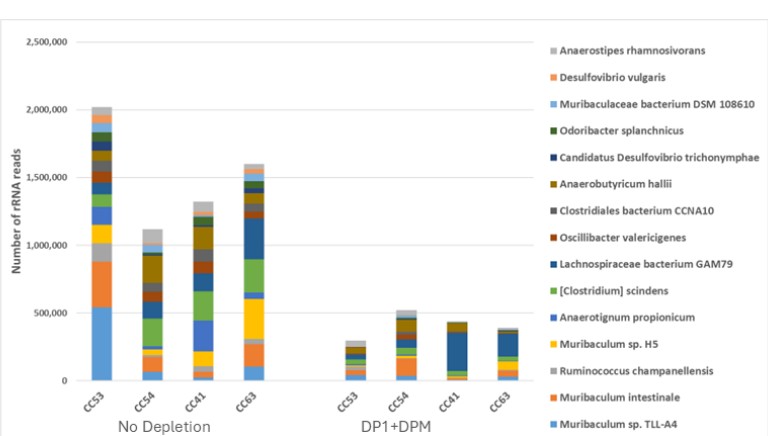

C

**FIG 1** Depletion of rRNA from mouse cecal, ileum, and liver samples for RNA-Seq library preparation and sequencing. (A) Summary of library preparation and sequencing. Total RNA samples were rRNA depleted and converted into stranded Total RNA libraries and sequenced. The resulting raw FASTQ files were quality

Fig 1 (Continued)

filtered with Trimmomatic to remove any reads that were less than 50 bases and below Q20. The remaining reads were then aligned with BBDUK to rRNA sequences using the Silva database as reference. The detected rRNA reads were collected as a new FASTQ file that was used for taxonomic analysis to determine the proportion of species contributing to the rRNA contamination. Reads filtered for rRNA were further processed for MetaT analysis using Humann3 to assign gene families and metabolic pathways. (B) Analysis of host and rRNA content of 56 cecal, 62 ileum, and 62 liver samples (average of two replicates shown, see File S1 for ordered list of sample names). Liver and ileum samples are dominated by host transcriptome (red shading, mm9 genome reference), while cecal samples are primarily metatranscriptome reads (blue shading = % retained; the proportion of reads that are not host or rRNA used for MetaT analysis). LSU and SSU refer to the rRNA large subunit and small subunit, respectively, for each species type listed. Host rRNA refers to reads that align to mouse rRNA. Rfam refers to other RNA family types, like tRNA. The asterisks at the top indicate the samples chosen for probe design in Fig. 2 (CC6, CC7, CC9, CC51, CC53, CC54, CC58). (C) Taxonomic analysis of the collected rRNA reads from either undepleted (left side) or samples depleted (right side) with DP1 and DPM indicates that no species dominate the remaining rRNA content of the samples following depletion. Samples CC53, CC54, CC41, CC63 shown (see Table S1). The asterisks at the top indicate two of the samples chosen for probe design in Fig. 2.

As previously established (25), following alignment to the rRNA database, the short sequencing reads tend to cluster into longer stretches, or regions, of rRNAs. We therefore limited the number of additional mouse-specific probes by altering two main parameters of the design process: the number of abundant rRNA regions per sample used for designs and the spacing of the probes across the regions of interest. The region numbers and spacing option strategy are summarized in Fig. 2A. We chose 7 of the 56 cecal samples demonstrating high rRNA levels (~32–55% of total reads) and subjected them to further analysis. The abundant rRNA regions were collected from each of these samples and ranked according to coverage depth. For each of the seven individual samples, we categorized and collected the top 50, 30, 25, or 20 most abundant regions, and then for each category, the samples were combined. To limit redundant sequences, the regions within each category were subjected to pairwise alignment. If any two sets of regions demonstrated at least 80% sequence identity, only one was randomly selected for further processing. Within the remaining regions, 50 nucleotide antisense probes were designed; however, the gap between adjacent probes was altered from 25 to 50 nucleotides apart in five-nucleotide increments. The expectation is that probes designed against a larger number of regions (i.e., Top 50 vs Top 20) and spaced closer together (i.e., Gap 25 vs Gap 50) will inherently result in a larger number of probes than selecting a smaller number of regions and probes spaced further apart. The results of the number of probes designed per category from this pipeline are shown in Fig. 2B and demonstrate the expected trends. For example, designs based upon the Top 50 regions and spaced 25 nucleotides apart result in 492 probes, but if spaced apart by 50 nucleotides, it results in 380 probes. In addition, comparing the use of more vs less abundant regions within each spacing category also demonstrates the expected trends.

While it is not practical to test every probe design option, we chose to continue with two probe designs that essentially fall at each end of the design spectrum. Furthermore, considering the cost and convenience of probe synthesis, we wanted to limit the number of probes to <384 oligonucleotides. For these reasons, we chose to move forward with the probe designs for the Top 50 regions, spaced 50 nucleotides apart (the 5050 probe set; 380 oligos) and the designs for the Top 20 spaced 25 nucleotides apart (the 2025 probe set; 317 oligos). These probe sets were synthesized, combined with both DP1 and DPM, and tested for rRNA depletion efficiency in a subset of the cecal samples used in the above experiments.

## Improved efficiency of rRNA depletion using mouse-specific supplementary probes

Fifteen samples were chosen from the initial 56 tested previously, including six taken from the learning set used for probe design, and were subjected to three different depletion options: depleted solely with DP1 and DPM, or depleted with DP1 and DPM plus the 5050 or 2025 probe sets (Fig. 3A). In 15/15 samples, the addition of the extra probes resulted in a reduction of rRNA reads, regardless of whether the samples were used for probe design or not. Furthermore, for the majority of samples (9/15), the 2025

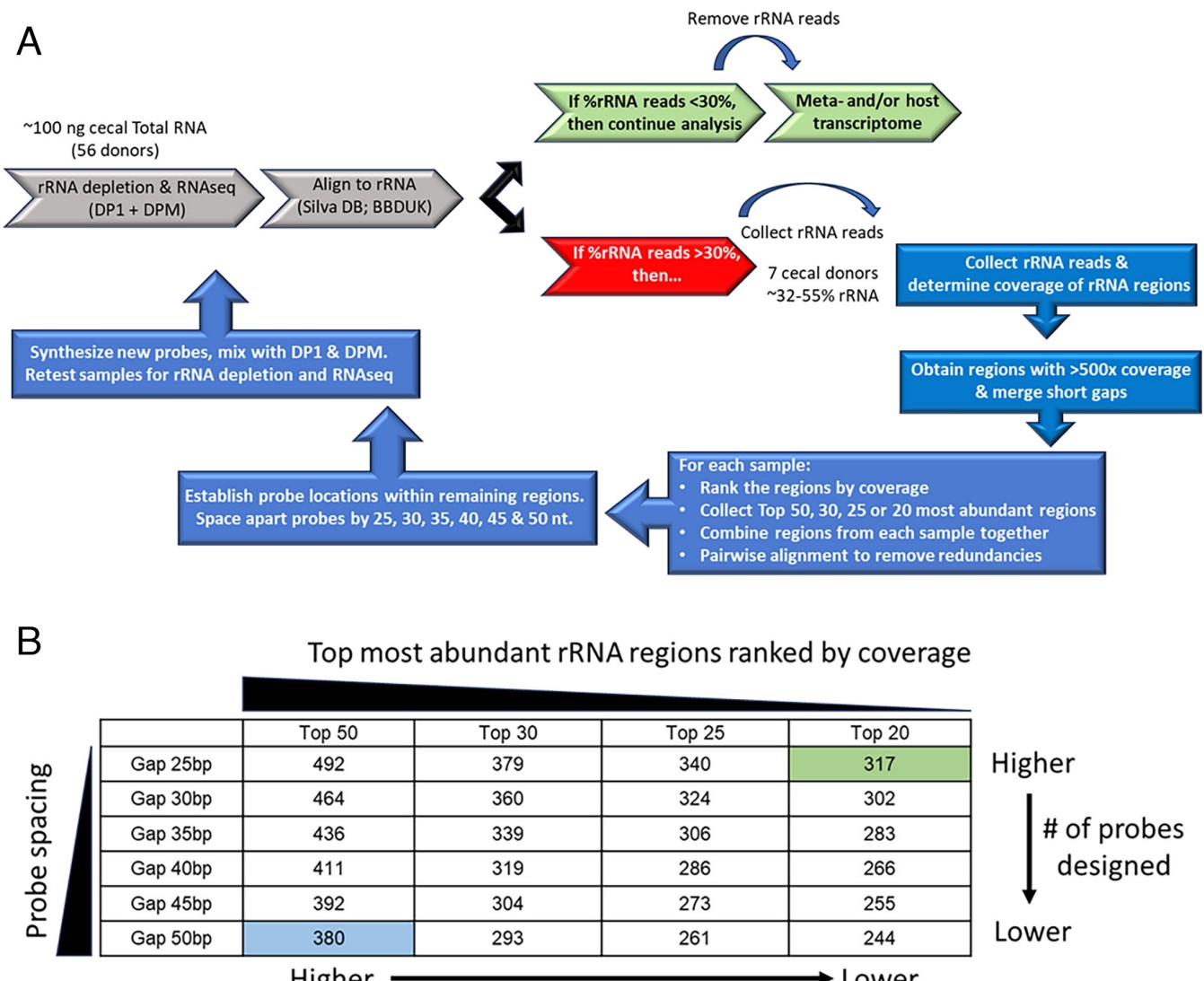

**FIG 2** Probe design strategy based on the most abundant reads. (A) From the 56 cecal samples that were initially processed, seven samples with %rRNA reads >30% were chosen as the learning set for additional probe design. For each sample, the rRNA reads were collected and aligned to the Silva database to determine the coverage depth across regions of rRNA transcripts. Any regions that were covered >500× were collected and ranked from highest to lowest. Two filtering options were used to minimize the number of probes needed. First, the regions that make up the top 50, 30, 25, or 20 most abundant sequences were binned, and the equivalent regions from each sample were combined. Second, for each combined set of regions, antisense probes were designed with various spacing options ranging from 25 to 50 nt apart. The probes can then be ordered, combined with DP1 and DPM, and the Total RNA samples retested for rRNA depletion and overall RNA-Seq performance. (B) The number of probe designs relative to the two main criteria utilized, the number of abundant regions used, and the spacing of the probes across the target regions. Two pools were chosen for synthesis and further analysis. The 5050 pool is composed of 380 probes designed against the top 50 most abundant regions and spaced 50 nt apart. The 2025 pool of 317 probes is designed to target the top 20 most abundant regions and is spaced 25 nt apart.

probe set demonstrated better depletion than the 5050 set. This demonstrates that the addition of either of the new probe sets improves rRNA depletion performance across multiple samples/donors. Perhaps a more practical measure of the performance of the added probes is the percentage of retained reads that are ultimately used for MetaT analysis. The proportion of retained reads from the samples with or without the supplemental probes is summarized in Fig. 3B. Depletion with DP1 and DPM results in a sample mean of ~60% retained but also indicates a relatively broad range from ~40%–90%. Addition of the 2025 probes significantly increased the average percent retained

A

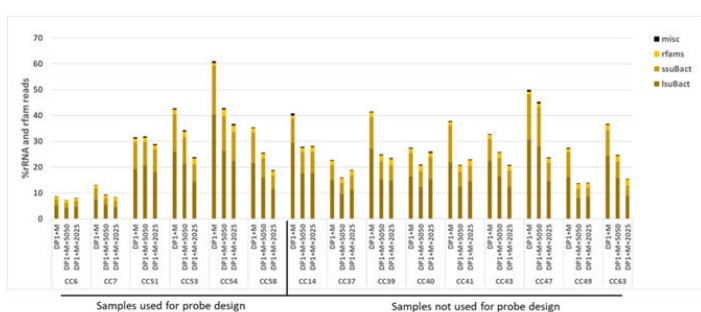

B

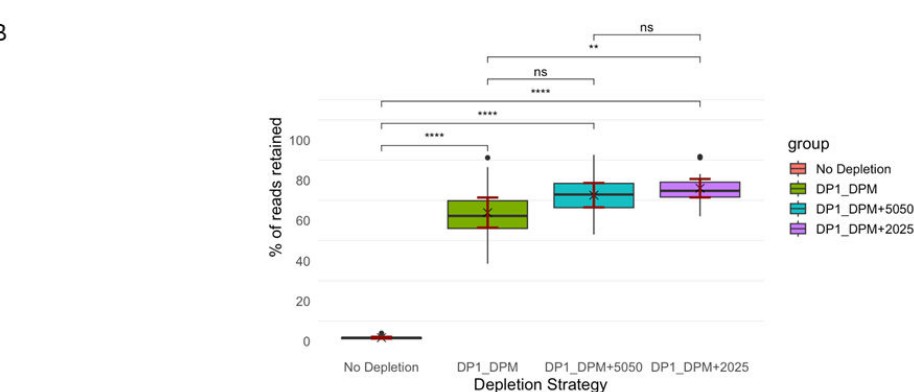

C

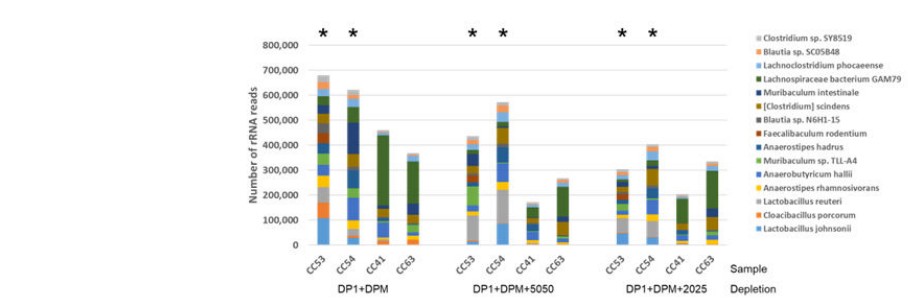

D

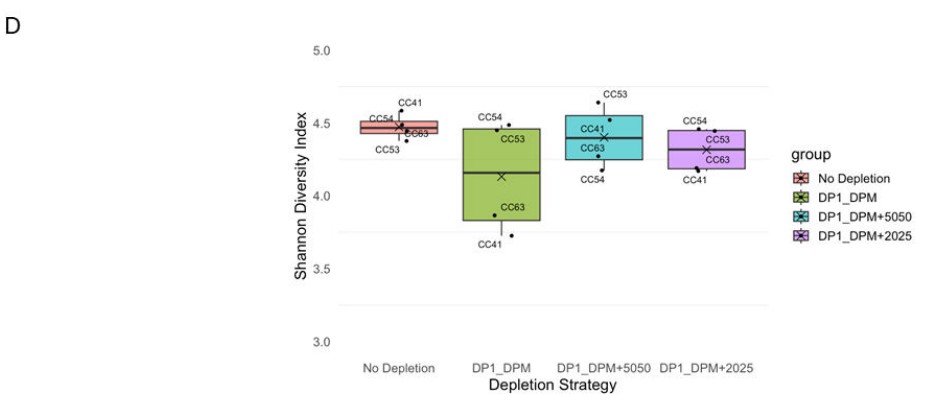

**FIG 3** Inclusion of supplementary probes improves rRNA depletion and increases the proportion of reads available for MetaT analysis. (A) Percentage of rRNA reads in samples depleted using standard probes (DP1 + DPM) or with supplementary probes 5050 or 2025 included. Samples on the left were used for supplementary probe designs, and samples on the right were not. In

Fig 3 (Continued)

both cases, the majority of samples (14/15) demonstrate a decrease in the rRNA reads upon inclusion of the additional probes. Inclusion of the 2025 probe set results in the least amount of rRNA reads. (B) The inclusion of the 2025 probes both increases the percentage of reads available for MetaT analysis by ~15% and provides greater consistency between individual donors. Pairwise comparisons were conducted using two-sided *t*-tests, with significance annotated as ****$P \leq 0.0001$; **$P \leq 0.01$; and ns, not significant. Exact *P*-values for each comparison are provided in Table S2. The 95% confidence intervals are represented by the red lines above and below the box plots. (C) Taxonomic representation of rRNA reads remaining following depletion conditions determined using the Illumina DRAGEN Metagenomics App on BaseSpace, comparing the standard probes alone (left, DP1 + DPM) or when the 5050 (middle) or 2025 (right) supplementary probes are included. Four representative samples are shown; two of the samples (CC53 and CC54) were used for probe design, and two samples (CC41 and CC63) were not. The results indicate that the newly designed probes do not result in targeting the rRNA from only a few specific taxa but are more dispersed across several taxa. Only the top 15 most abundant species (sorted by sample CC53-DP1 + DPM) are shown. A more comprehensive analysis of the top 100 species is shown in Fig. S2. (D) rRNA reads were classified taxonomically using the Illumina DRAGEN Metagenomics App in BaseSpace, and the resulting species-level profiles were used to calculate the Shannon diversity index for each sample. All depletion methods retained a taxonomically diverse set of rRNA reads (Shannon index > 3.7), except for the two lowest data points in the DP1 + DPM condition, representing samples CC41 and CC63. Upon inclusion of the additional probes, the diversity of these two samples improves. This suggests that the depletion strategies do not introduce strong taxonomic bias in the remaining rRNA.

to ~75% ($P < 0.01$; see Table S2), but also improved the consistency of depletion between samples as well. The range of the percentage retained narrowed to ~60%–90%. The 5050 probe set also improved the amount of retained reads recovered, but was not statistically significant compared to depletion with DP1 and DPM (Table S2), indicating that the 2025 probes demonstrate better performance in these samples. These results show that the 2025 probe design strategy was able to both improve the efficiency (average % mRNA reads) and consistency (spread between samples) of rRNA depletion from the cecal samples, resulting in an average recovery of ~15% more reads for functional data analysis.

To understand how the addition of the extra probe sets could impact the taxonomic distribution of rRNA reads, we collected the remaining rRNA content from representative samples, including two of the samples used for probe design and two that were not, and performed taxonomic analysis (Fig. 3C). The purpose was to determine if any of the leftover rRNA reads are overtly dominated by a particular species in the samples following depletion, which might suggest the probe design strategy somehow missed certain abundant sequences. As was observed previously (Fig. 1C), in the samples used for probe design, CC53 and CC54, no particular species substantially dominate the rRNA content following depletion with DP1 and DPM (Fig. 3C). Conversely, the remaining rRNA content of the two samples not used for probe design (CC41 and CC63) is less evenly distributed and contains a larger proportion of *Lachnospiraceae*. Addition of either the 5050 or 2025 probe sets does not appear to bias the representation of any particular genus or species in the overall taxonomic profile of the samples, generally compressing the proportion of reads assigned to all of them. However, one noticeable exception is *Lachnospiraceae* in samples CC41 and CC63. Despite these samples not being part of the training set, the addition of the extra probes did provide a change in the proportion of reads assigned to this species. CC41 shows a major reduction in *Lachnospiraceae* rRNA reads following the addition of the 5050 probe set and a more modest depletion upon the addition of the 2025 probes. This suggests that even though these two samples were not used for the learning set, the generated probes are still effective at removing rRNA content not specifically included in the design. These visual observations are also consistent with Shannon diversity analysis (Fig. 3D), where inclusion of the additional probes improves the Shannon diversity index of CC41 and CC63. Furthermore, the diversity index of the samples depleted with the additional probes is similar to that of the No Depletion conditions. These results demonstrate that this probe design strategy can provide depletion outside the taxonomic range of the samples used in the training set

and does not bias taxa representation in the depleted samples, reinforcing the value of this strictly abundance-based and species-agnostic approach to the design method.

## Analysis of off-target effects from additional probes

An important consideration when supplementing additional probes for rRNA depletion is whether there are any off-target or otherwise non-specific effects of the probes on the remaining content that could impact the ability to perform either host transcriptome or MetaT analysis. Any of the additional probes designed against the most abundant sequences could potentially hybridize host or microbial mRNAs and result in depleting those regions of the transcript and potentially causing erroneous gene expression profiling results. To investigate this possibility, we first analyzed the mouse transcriptome by gene expression comparisons of host transcripts with or without the additional probes (Fig. 4A and B). The cecal samples contain very little host transcriptome (Fig. 1B), so we utilized a subset of the liver samples to perform differential gene expression analysis of the host transcriptome, comparing depletion with only DP1 and DPM vs the inclusion of the additional probes (Fig. 4A and B). In both cases, the addition of the supplemental probes for depletion does not notably impact host gene expression in the mouse liver samples, with a Pearson $R^2$ correlation of 0.96 when adding either the 5050 or 2025 probe sets compared to just DP1 and DPM (Fig. 4A and B). We also compared gene family relative abundance within the metatranscriptome (Fig. 4C through H) with or without the addition of the probe sets. To examine any possible detrimental effects the additional probes may have on the MetaT results, we performed a rank correlation analysis of the cecal samples from Fig. 3, specifically focusing on the correlation between the relative abundance of gene pathways. Two samples were chosen for this analysis: CC53 (used for probe design; Fig. 4C through E) and CC41 (used for probe design; Fig. 4F through H). Spearman's rank correlation ($\rho$) demonstrates pairwise comparisons with quite good concordance between probe depletion strategies, where $\rho$ values range from ~0.92 to 0.95. This indicates that the inclusion of the additional probes has no obvious detrimental effects on MetaT analysis. Together, the results suggest that rRNA depletion using either the 5050 or 2025 probe set causes minimal bias for transcriptome profiling of these complex sample types.

## DISCUSSION

In this work, we describe a novel strategy to adapt and optimize a probe pool initially created for the enzymatic depletion of abundant rRNA sequences in human gut microbiome samples to enable effective and consistent use in mouse cecal samples. Efficient depletion of rRNA sequences in these sample types greatly increases the informational content for use in MetaT analysis. The main goal in these types of studies is to determine the gene expression profile and functional/metabolic characteristics of the microorganisms residing in the host. Knowledge about host-microbe interplay relies on understanding both microbiome contribution as well as the host response and how it can vary with specific gene expression contributions from the microbial community. At our current level of understanding, it is not clear to what extent the host response is dependent on specific microorganisms being present. Thus, being able to compare the microbiome population composition to the metatranscriptomic profile during controlled stimulation experiments is critical.

The approach to probe design used in this work is agnostic toward the taxonomic content of the samples and instead relies on collecting abundant microbial rRNA sequences from several mouse cecal donors, setting coverage thresholds, and varying probe spacing options to efficiently create additional probes optimized for mouse cecal microbial rRNA content. We initially tested 56 individual cecal donors using the RZPM probe sets (DP1 and DPM) designed against human stool and determined that for many donors, the rRNA depletion resulted in <30% of the sequencing reads matching microbial rRNA. However, several donors demonstrated higher rRNA content: in some cases, up to ~50% of reads. We chose seven donors with rRNA content >30% for use as a training

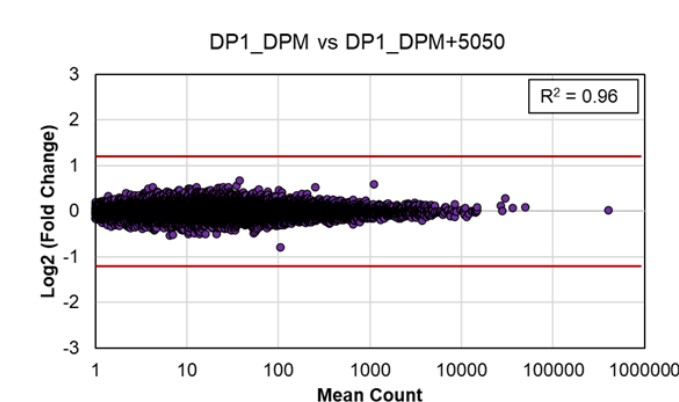

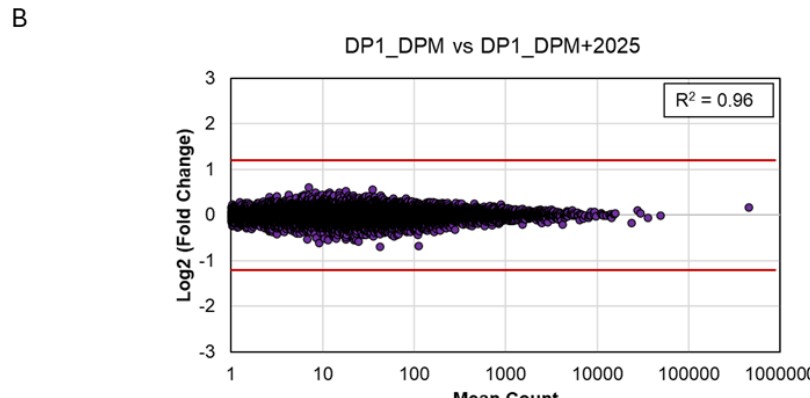

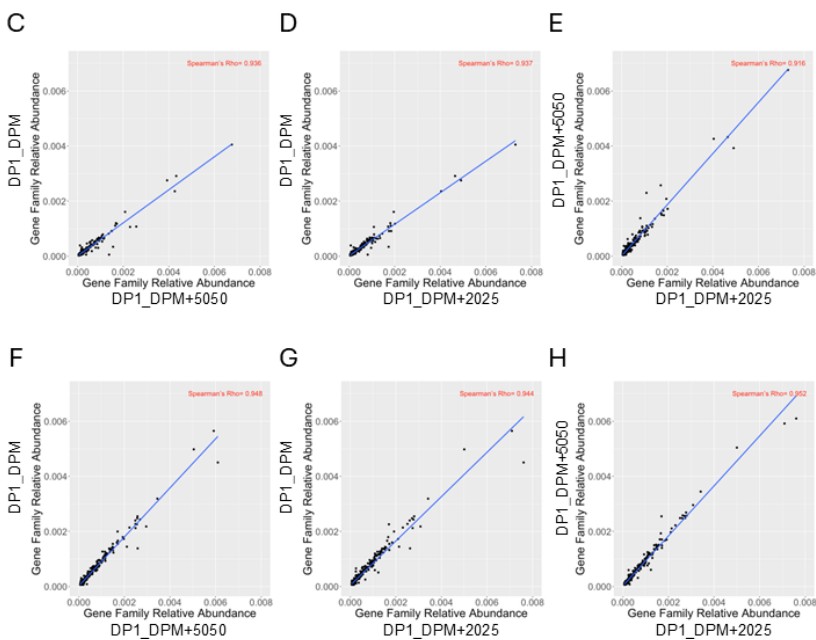

**FIG 4** Inclusion of supplementary probes does not result in significant bias to MetaT analysis. (A and B) Gene expression comparison of liver samples from three mice treated as replicates (LVR20, LVR25, LVR49) comparing depletion conditions using the standard probes (DP1_DPM) vs inclusion of the 5050 (A) or 2025 (B) probe sets. In both cases, the Pearson $R^2$ values of 0.96 (Continued on next page)

Fig 4 (Continued)

suggest no significant bias is introduced to the host transcriptome upon inclusion of the supplementary probes. (C–H) Scatter plots of normalized gene family abundances, where each point corresponds to a gene family, comparing depletion conditions in either a sample used for probe design (CC53; C–E) or a sample that was not (CC41; F–H). The X and Y axes show the fraction of total abundance contributed by that gene family in each sample. The Spearman's rank coefficient, or Rho ($\rho$), indicates how similarly the gene families are distributed between the two samples. Higher values suggest a stronger correlation. In all comparisons, $\rho$ demonstrates strong agreement (>0.91), suggesting minimal bias is introduced to the metatranscriptomes by inclusion of the supplementary probes.

set for additional probe design. Abundant rRNA sequences were collected from these individual samples and merged to create abundant rRNA "regions" and then ranked by median coverage depth from highest to lowest. To investigate probe design efficiency, we combined two options. First, we filtered sequences based upon specific coverage thresholds, and second, designed anti-sense probes with various spacing across the rRNA regions selected by coverage. This matrix of options produced probe design pools with various numbers of individual oligos that trend as generally expected; designs against higher numbers of abundant regions and spaced closer together generate more probes than using fewer regions and spacing the probes further apart. We then chose two design pools that represent compromises between these two extremes (the green and blue cells in Fig. 2B) and tested their performance on both the same cecal samples used for training as well as samples that were not. Both design options demonstrate improved rRNA depletion performance, but the more "packed" version (using the top 20 most abundant regions and probes spaced 25 nt apart; named the 2025 probe set) results in a statistically significant increase in the percentage of non-rRNA reads generated for MetaT analysis. In addition, the ranges of rRNA content from different donors are generally more compact and consistent when compared to the use of the human gut-centric probes by themselves.

The microbiome is present virtually throughout the human body (30). However, most areas of the human biome in healthy individuals contain only trace amounts of microbes, with the bulk present within the digestive and respiratory tracts, which provide enriched conditions for their growth due to contact with the external environment (31). This knowledge has resulted in a great deal of investigation into gut microbiomes and the role they play in the health of their hosts, as well as the effect of the host's diet and health on microbial composition and gene expression. Several studies have suggested an interplay between microbiome metabolic activity and the response of the host to various interventions. The diet of the host is perhaps an obvious example where consumption of certain nutrients clearly results in changes to the gut flora and activity (2, 32). Therefore, it is important to consider the ability to monitor both host and microbe responses to aspects such as dietary stress, disease states, or even medicinal treatment. With this in mind, we performed rRNA depletion and RNA-Seq analysis of three distinct mouse microbiome types: cecal, terminal ileum, and liver. The proportional amount of microbial vs host transcriptome varies considerably between these sample types. The liver and terminal ileum samples contain predominantly host RNA, while the cecal content is mostly microbial. The existing RZPM kit is designed to remove the common rRNA transcripts from human, mouse, and rat samples via the use of the DP1 probe pool. Furthermore, previous work established the use of DPM for microbial rRNA removal from human stool, vaginal, and oral microbiomes. This provides an integrated tool to allow the simultaneous gene expression profiling of both the host and human microbiome. A major goal of this current work is to provide tools that will maximize the amount of useful MetaT information available using an optimized set of supplemental probe pools for mouse cecal content.

In addition to providing a manageable set of depletion probes, we demonstrate that the use of these supplemental probes does not introduce unwanted bias to either the host or the meta-transcriptome, in terms of microbial species or metabolic pathway gene expression. Our results suggest that the combination of the 2025 probe set with

the RZPM probes (DP1 and DPM) provides an effective tool to enable MetaT analysis of mouse cecal microbiomes. Furthermore, these additional probes are designed to limit the total number of probes to be cost-effective, adding an estimated $10 per sample to the total cost of library preparation and sequencing.

The ability to remove abundant unwanted sequences from total RNA is critical for any whole transcriptome RNA-Seq analysis (11). Sequencing total RNA without rRNA removal generally results in most sequencing reads aligning to rRNA. Microbiome samples are especially susceptible; typically, >95% of all reads are identified as microbial rRNA (Fig. 3). In some studies, this rRNA content has been used to track the taxonomic profile of microbiomes and rank species abundance based upon the proportion of reads assigned to each species. Much like metagenome analysis, the proportion of each species can then be used as a proxy to infer metabolic or functional activity of the population (33, 34). However, microbiomes are extremely complex systems, containing hundreds to thousands of different species, and the proportional existence of a particular taxon or species related to its metabolic contribution to enzymatic pathways is inferential at best. Abundant species may be relatively quiescent or vice versa, and small species populations may be extremely significant to the metabolic state of the population. Furthermore, microbiomes are typically highly dynamic environments/ecosystems that will rapidly adapt to changes in diet, health conditions, and the various medications used to treat or manage them. It is for these reasons that having the means to effectively profile microbial gene expression changes via MetaT analysis is critical to our understanding of microbiomes as well as their influence and interactions with their hosts.

## ACKNOWLEDGMENTS

A.Z. is a co-founder and acting CMO of Endure Biotherapeutics. He holds equity in the company.

A.Z. is supported by NIH R01 EB030134, R01 AI163483, U01 CA265719, and VA Merit BLR&D Award I01 BX005707. Some authors receive institutional support from NIH P30 DK120515, P30 DK063491, P30 CA014195, and UL1 TR001442.

## AUTHOR AFFILIATIONS

[1]Illumina Inc., San Diego, California, USA

[2]Division of Gastroenterology, University of California, San Diego, La Jolla, California, USA

[3]Department of Neurosciences, University of California San Diego, La Jolla, California, USA

[4]Departments of Neurosciences and Pathology, University of California San Diego, La Jolla, California, USA

[5]Shu Chien-Gene Lay Department of Bioengineering, University of California San Diego, La Jolla, California, USA

[6]Center for Microbiome Innovation, University of California San Diego, La Jolla, California, USA

[7]Division of Gastroenterology, Jennifer Moreno Department of Veterans Affairs Medical Center, La Jolla, California, USA

[8]Retired Researcher, La Concha Pass, Austin, Texas, USA

## AUTHOR ORCIDs

Amir Zarrinpar http://orcid.org/0000-0001-6423-5982
Scott Kuersten http://orcid.org/0009-0007-5058-4920

## AUTHOR CONTRIBUTIONS

Morgan Roos, Data curation, Formal analysis, Investigation, Methodology, Visualization, Writing – review and editing | Samuel Bunga, Data curation, Methodology, Software, Writing – review and editing | Asako Tan, Conceptualization, Investigation, Methodology, Software, Writing – review and editing | Erica Maissy, Investigation, Methodology, Resources, Writing – review and editing | Dylan Skola, Methodology, Software,

Visualization, Writing – review and editing | Alexander Richter, Project administration, Supervision, Writing – review and editing | Daniel S. Whittaker, Resources, Writing – review and editing | Paula Desplats, Resources, Writing – review and editing | Amir Zarrinpar, Resources, Supervision, Writing – review and editing | Rick Conrad, Writing – original draft, Writing – review and editing | Scott Kuersten, Conceptualization, Formal analysis, Methodology, Project administration, Resources, Supervision, Visualization, Writing – original draft, Writing – review and editing

## DATA AVAILABILITY

The FASTQ files obtained for this work are available via the SRA under BioProject accession ID PRJNA1258316.

## ETHICS APPROVAL

All animal research was performed in accordance with the University of California, San Diego, Institutional Animal Care and Use Committee.

## ADDITIONAL FILES

The following material is available online.

### Supplemental Material

**File S1 (mSystems00167-25-s0001.xlsx).** List of rRNA depletion probes.
**Supplemental material (mSystems00167-25-s0002.pdf).** Supplemental figures and tables.

### Open Peer Review

**PEER REVIEW HISTORY (review-history.pdf).** An accounting of the reviewer comments and feedback.

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
