## [Reviewer comments · mSystems]

Optimizing mouse metatranscriptome profiling by selective removal of redundant nucleic acid sequences

Scott Kuersten, Morgan Roos, Samuel Bunga, Asako Tan, Erica Maissy, Dylan Skola, R Richter, Daniel Whittaker, Paula Desplats, Amir Zarrinpar, and Rick Conrad

Corresponding Author(s): Scott Kuersten, Illumina Inc

Review Timeline:

Submission Date:	February 4, 2025
Editorial Decision:	April 7, 2025
Revision Received:	May 16, 2025
Accepted:	May 20, 2025

Editor: Aaron Miller

Reviewer(s): Disclosure of reviewer identity is with reference to reviewer comments included in decision letter(s). The following individuals involved in review of your submission have agreed to reveal their identity: Mangesh Suryavanshi (Reviewer #2)

Transaction Report:

DOI: <https://doi.org/10.1128/msystems.00167-25>

Re: mSystems00167-25 (Optimizing mouse metatranscriptome profiling by selective removal of redundant nucleic acid sequences)

Dear Dr. Scott Kuersten:

Revision Guidelines

Sincerely,
Aaron Miller
Editor
mSystems

Reviewer #1 (Comments for the Author):

Ross et al. report a new assay to improve rRNA depletion in metatranscriptomic analyses using mouse samples. Since several microbiome studies employ murine models, there is great need for dedicated strategies to remove rRNA from mouse samples. The authors successfully improved probe design and demonstrated that their new approach significantly upgraded the quality of the RNA-seq data and the amount of information.

Reviewer #2 (Comments for the Author):

For Abstract Section

Question 1:

Can you explain what you mean by 'rational probe design'? Why is it based on sequence abundance instead of taxonomy?

Introduction Section

Question 2:

Could you include some recent (2023-2025) studies on rRNA depletion or metatranscriptomics in mouse models?

Materials and Methods Section

Question 3:

Did you use RNase inhibitors during sample handling to protect RNA quality?

Question 4:

Why did you choose 1 μL of 1 pmol/ μL probe per reaction? A quick justification or reference would be helpful.

Question 5:

Could you add a supplementary table listing all software versions and settings used? Also, will raw data and scripts be shared (e.g., GEO, GitHub)?

Results Section

Question 6:

Were technical replicates done for RNA extraction or depletion steps? This might help explain sample variability.

Question 7:

In Figure 3B, can you add 95% confidence intervals along with p-values?

Question 8:

Have you thought about including Shannon diversity indices to show that no species dominates residual rRNA?

Probe Design and Testing Section

Question 9:

Did you apply any filters like GC content or melting temperature when designing the probes?

Question 10:

Could you explain the '5050' and '2025' probe naming earlier in the results section?

Question 11:

How did you choose the validation samples? Randomly or based on % rRNA? Any technical replicates?

Question 12:

Were batch effects (e.g. from probe synthesis or sequencing) checked or controlled?

Discussion Section

Question 13:

Can you give an idea of how much the custom probes add to the cost per sample? if possible.

Question 14:

Where will you share the probe sequences, FASTQ files, and analysis results-GEO, SRA, GitHub?

Figures and Supplementary Data Section

Question 15:

For the stacked bar plots, clearer legends and taxa labels would really help.

Question 16:

In Supplementary Table 2, adding effect sizes along with p-values would show the actual impact better.

Question 17:

In Supplementary Figures 1 & 2, consider writing full species names in the legends instead of using abbreviations.

Question 18:

There are a few small typos (check out like: c ost-effective). A quick proofread should catch these.

Response to Reviewers: Roos et al. (mSystems00167-25)

Reviewer #1 (Comments for the Author):

Roos et al. report a new assay to improve rRNA depletion in metatranscriptomic analyses using mouse samples. Since several microbiome studies employ murine models, there is great need for dedicated strategies to remove rRNA from mouse samples. The authors successfully improved probe design and demonstrated that their new approach significantly upgraded the quality of the RNA-seq data and the amount of information.

We thank reviewer 1 for their comments.

Reviewer #2 (Comments for the Author):

For Abstract Section

Question 1:

Can you explain what you mean by 'rational probe design?' Why is it based on sequence abundance instead of taxonomy?

We added additional text to the introduction section to explain that probe design based upon taxonomic content would require too many probes for the depletion method to process. Also, many of them would not actually be required due to inherent limitations of conversion of the rRNA into cDNA. However, probe design based upon the most highly abundant sequences help to limit the overall number of probes and the cost of the assay.

Introduction Section

Question 2:

Could you include some recent (2023-2025) studies on rRNA depletion or metatranscriptomics in mouse models?

We have added 5 additional references that include both rRNA depletion techniques for complex bacterial samples and metatranscriptome studies, including specifically for murine microbiomes.

Question 3:

Did you use RNase inhibitors during sample handling to protect RNA quality?

RNase inhibitors are typically not needed if the RNA extractions are performed effectively. That said, an RNase inhibitor is embedded in the first strand synthesis reagent provided in the Illumina Stranded Total RNA Prep with Ribo-Zero Plus Microbiome kit.

Question 4:

Why did you choose 1 μL of 1 pmol/ μL probe per reaction? A quick justification or reference would be helpful.

We added additional text in the method section to explain that this concentration was used because it matches the probe concentrations in the Ribo-Zero Plus® Microbiome kit.

Question 5:

Could you add a supplementary table listing all software versions and settings used? Also, will raw data and scripts be shared (e.g., GEO, GitHub)?

We have added Supplementary Table 3 listing all the BSSH Apps used for the RNAseq analysis. FASTQ files are uploaded to SRA (description provided in Materials and Methods section). We also added a comment stating the scripts for probe design will be provided upon request, as we are currently preparing a separate manuscript focused on the computational approach of the probe design strategy.

Results Section

Question 6:

Were technical replicates done for RNA extraction or depletion steps? This might help explain sample variability.

No replicates were collected for extractions as this was a fairly large number of individual mouse donors (~60). For Figure 1B, two replicate RNAseq libraries were constructed and the results averaged, however for subsequent testing, where the same samples had to be tested several times with different depletions conditions, we were unable to perform replicates. This was simply due to sample volume limitations and the desire to retain residual sample quantity for the main study on Alzheimer disease mentioned in the introduction.

Question 7:

In Figure 3B, can you add 95% confidence intervals along with p-values?

We have updated Figure 3B by including the statistical significance between each comparison as well as the 95% confidence intervals for each group. The exact p-values for each comparison are still included in Supplementary Table 2.

Question 8:

Have you thought about including Shannon diversity indices to show that no species dominates residual rRNA?

We have added Figure 3D and updated the results section to highlight the Shannon Diversity Indices of the depletion conditions. These results suggest that addition of the supplemental probes improves the taxa diversity of the samples, reinforcing the more visual representation in Figure 3C.

Probe Design and Testing Section

Question 9:

Did you apply any filters like GC content or melting temperature when designing the probes?

No to both. All the probes (including the existing probes included in the RZPM kit) are 50 nucleotides in length, which from previous product development data and experience are sufficiently long enough to handle the assay conditions. Furthermore, since rRNA are known to have high GC content, adding filters for high GC sequences would likely introduce bias and/or lower the effectiveness of the assay. The probes are also designed to tile across the target sequences, so if specific individual probes are less efficient the adjacent probes can compensate for depletion activity.

Question 10:

Could you explain the '5050' and '2025' probe naming earlier in the results section?

It is unclear to us where to explain the naming strategy earlier in the results section without inherently adding significant amounts of additional text and risk being redundant with later comments.

Question 11:

How did you choose the validation samples? Randomly or based on % rRNA? Any technical replicates?

Samples used for probe design had %rRNA >30% and were also retested for validation of new probes. The validation samples that were not used for probe design were chosen somewhat randomly, but also based upon sample availability, which was inherently limited. Likewise, technical replicates were not done due to limited supply of RNA samples. Instead, we relied on the overall trend across samples demonstrating that adding the new probes reduced rRNA content for every sample tested (15/15) (Figure 3A and stated in the results section).

Question 12:

Were batch effects (e.g. from probe synthesis or sequencing) checked or controlled?

To limit the expense of probe synthesis, we did not have multiple batches of probes made by IDT. Additionally, based upon experience with the product development of the RiboZero Plus and RiboZero Plus Microbiome kits at Illumina, testing multiple lots of probe synthesis is not typically needed. This is because IDT performs detailed QC on each synthesized probe and provides both quantity and mass spectroscopy data to ensure integrity of the oligo pools used in this work.

Discussion Section

Question 13:

Can you give an idea of how much the custom probes add to the cost per sample? If possible.

We added a sentence in the Discussion section that reads, "Furthermore, these additional probes are designed to limit the total number of probes to be cost effective, adding an estimated \$10 per sample to the total cost of library preparation and sequencing."

Question 14:

Where will you share the probe sequences, FASTQ files, and analysis results-GEO, SRA, GitHub?

Probe sequences are included in the Supplementary file attached with the revised manuscript. FASTQ files are uploaded to SRA and found via the BioProject accession ID PRJNA1258316. A related comment was added to the material and method section. For the scripts used for probe design, we added a comment to the materials and methods section stating that the scripts can be made available upon request. We are preparing a separate manuscript focused on the computational use and testing of these scripts for the probe design strategy.

Figures and Supplementary Data Section

Question 15:

For the stacked bar plots, clearer legends and taxa labels would really help.

We have increased the font size of the legends to make the genus and species names more obvious. We could also consider removing these figures. Since they are for the top 100 species, they are visually quite complex and may not be needed to make the point. We chose to include them as supplementary data to emphasize the complex diversity of the samples and the unbiased nature of the depletion strategy.

Question 16:

In Supplementary Table 2, adding effect sizes along with p-values would show the actual impact better.

We have added Cohen's d values to calculate the effect sizes to Supplementary Table 2 to better demonstrate the magnitude of differences between groups. These results indicate no large differences in effect sizes between comparison groups.

Question 17:

In Supplementary Figures 1 & 2, consider writing full species names in the legends instead of using abbreviations.

The genus and species names that are listed in the figure legends are based upon the Kraken 2020 reference database that is part of the DRAGEN Metagenomics App in BaseSpace.

Question 18:

There are a few small typos (check out like: c ost-effective). A quick proofread should catch these.

We performed a spell check to correct the typos.

Re: mSystems00167-25R1 (Optimizing mouse metatranscriptome profiling by selective removal of redundant nucleic acid sequences)

Dear Dr. Scott Kuersten:

Your manuscript has been accepted, and I am forwarding it to the ASM production staff for publication. Your paper will first be checked to make sure all elements meet the technical requirements. ASM staff will contact you if anything needs to be revised before copyediting and production can begin. Otherwise, you will be notified when your proofs are ready to be viewed.

Sincerely,
Aaron Miller
Editor
mSystems

Reviewer #1 (Comments for the Author):

Roos et al. carefully addressed the concerns of reviewer 2 and included all necessary modifications. The paper is ready to be accepted.

Reviewer #2 (Comments for the Author):

Authors did very nice study and cleared all raised concerns in revised draft.